# Decadal Scale Variability of Larsen Ice Shelf Melt Captured by Antarctic Peninsula Ice Core

**B. Daniel Emanuelsson, Elizabeth R. Thomas \*, Jack D. Humby and Diana O. Vladimirova**

Ice Dynamics and Paleoclimate Group, British Antarctic Survey, Natural Environment Research Council, High Cross, Madingley Road, Cambridge CB3 0ET, UK

\* Correspondence: lith@bas.ac.uk

**Abstract:** In this study, we used the stable water isotope record ($\delta^{18}O$) from an ice core drilled in Palmer Land, southern Antarctic Peninsula (AP). Utilizing $\delta^{18}O$ we identified two climate regimes during the satellite era. During the 1979–1998 positive interdecadal Pacific oscillation (IPO) phase, a low-pressure system north of the Weddell Sea drove southeasterly winds that are associated with an increase in warm air mass intrusion onto the Larsen shelves, which melted and a decreased sea ice concentration in the Weddell Sea/increase in the Bellingshausen Sea. This climate setting is associated with anomaly low $\delta^{18}O$ values (compared with the latter IPO period). There is significantly more melt along the northern AP ice shelf margins and on the Larsen D and southern Larsen C during the 1979–1998 IPO positive phase. The IPO positive climatic setting was coincidental with the Larsen A ice shelf collapse. In contrast, during the IPO negative phase (1999–2011), northerly winds caused a reduction in sea ice in the Bellingshausen Sea/Drake Passage region. Moreover, a Southern Ocean north of the Weddell Sea high-pressure system caused low-latitude warm humid air over the tip and east of the AP, a setting that is associated with increased northern AP snowfall, a high $\delta^{18}O$ anomaly, and less prone to Larsen ice shelf melt.

**Keywords:** ice shelf melt; decadal-scale variability; interdecadal Pacific oscillation (IPO); water stable isotopes; ice cores; Antarctic Peninsula

## 1. Introduction

There has been a mass loss from the Antarctic Peninsula (AP) ice sheet over recent decades [1]. The main contribution comes from basal melt on the western side of the peninsula at the Thwaites and Pine Island glaciers in the Amundsen Sea Embayment. The Larsen A and Prince Gustav ice shelves collapsed in 1995 [2] and Larsen B in 2002 [3]. The collapse of the Larsen ice shelves does not directly contribute to global sea-level rise but provides a buttressing effect for the continental AP ice sheet. That is, a speed-up of the ice sheet flow occurs with the disintegration of the ice shelves. Therefore, on the eastern AP, a mass loss followed the collapse of the Prince Gustav, Larsen A, and B ice shelves [4].

In recent years the strength of the westerly winds, the predominant wind direction, surrounding Antarctica has increased [5] (and references therein). For weak winds, the orography of the AP acts as an effective barrier. However, strong zonal winds can cross over the peninsula. These cross-over winds often emerge as warm and dry winds on the down-slope side on the eastern side of the mountain, this is called the foehn effect. Recent research has shown that foehn winds can cause melt on the Larsen ice shelves, even during the winter season [6]. Regions with low-firn air content (to the north and at the down-slope margins of AP) caused by melting on the Larsen C correspond to areas impacted by foehn winds and long melt durations [7].

Southern hemisphere and Antarctic climates exhibit strong decadal-scale variability driven by changes in remote teleconnection, associated with the interdecadal Pacific

oscillation (IPO) [8–10] and Indian Ocean Dipole (IOD) [11]. For example, England et al. 2014 [8] showed that the 1999–2011 warming hiatus corresponded with the negative IPO period. Moreover, Meehl et al. 2016 [10] linked the 2000–2014 Antarctic sea ice expansion to tropical Pacific decadal climate variability, the IPO. ENSO-related drivers have also been shown to bear a strong influence on Antarctic water-stable isotopes [12,13]. Goodwin et al. [14] examined large-scale atmospheric circulation modes in relation to the AP Bruce Plateau snow accumulation ice core record. They found a positive correlation between accumulation and SAM during the Pacific decadal oscillation's (PDO's) positive phase ~1975–2000. However, the correlation was negative or neutral during ~1945–1975 PDO's negative phase when SAM was negative and the westerlies weaker. The Bruce Plateau record only overlaps with a few years with the most recent PDO/IPO negative phase. Therefore, they could not examine possible changes during the ~1999 phase transition. With an examination period that extended beyond the satellite era, Porter et al. [15] demonstrated that the snow accumulation record from the Bruce Plateau ice core provides a robust sea ice extent proxy for the Bellingshausen Sea.

## 2. Materials and Methods

### 2.1. Drill Site

The Palmer drill site (73.86° S, 65.46° W, 1897 m a.s.l.) is located on the southern part of the AP, Palmer Land (Figure 1). The site is located on the AP divide with one side sloping towards the Amundsen–Bellingshausen Sea (ABS) and the other side slanting toward the Weddell Sea [16]. The core, firn, and ice were drilled in December 2012 to a depth of 133 m below the snow surface.

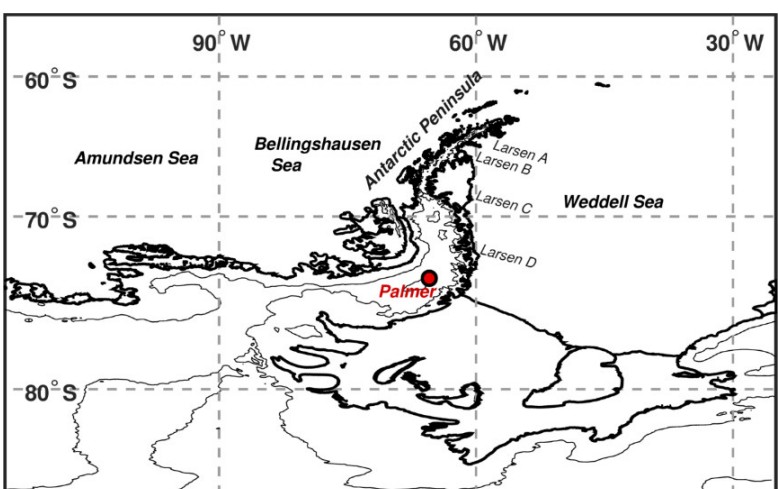

**Figure 1.** Antarctic Peninsula location map. The Palmer ice core site is indicated by a red dot. The location of the Larsen ice shelves and major Antarctic seas are shown. Contours (thin black lines) with 1000 m spacing indicate elevation above sea level.

### 2.2. Water Stable-Isotopes

The water isotopes (here we use $\delta^{18}O$) are reported as the ice core sample ratios $(H_2^{18}O/H_2^{16}O)_{sample}$ deviation from a reference ratio $(H_2^{18}O/H_2^{16}O)_{V-SMOW2/SLAP2}$ (Equation (1)). Three in-house standards were used for the daily calibrations (same method as in Keller et al. [17]) and they were calibrated using internationally recognized reference waters (V-SMOW2 and SLAP2 (Vienna-Standard Mean Ocean Water 2 and Standard Light Antarctic Precipitation 2)) [16].

$$\delta = 1000 \left( \frac{^iR}{^iR_{V-SMOW}} - 1 \right) \text{ (per mil, ‰),} \tag{1}$$

where $^iR$ is $^{18}O/^{16}O$ or $^2H/^1H$.

The measurements were performed on a laser spectroscopy instrument (Picarro, L2130-i, Santa Clara, CA, USA). The analyzer operates in a Continuous Flow Analysis (CFA) mode, where a continuous ice core sample stream is instantaneously evaporated and mixed with a dry carrier gas ($N_2$) in a custom-fabricated evaporation unit prior to being introduced to the analyzer, comparable to other similar setups [18–21].

The age scale for the Palmer ice core was established using annual-layer counting of multi-parameter high-resolution chemistry and isotope data [16]. The core spans the 1621–2011 age interval. Here we employed the 1979–2011 part of the Palmer $\delta^{18}O$ record, which overlaps with reliable reanalysis data (the satellite era). The Palmer $\delta^{18}O$ annual mean record is used and each annual mean encompasses isotope data from January–December.

### 2.3. Reanalysis Data

Annual means of monthly geopotential height at the 500-hPa level (Z500), meridional (V) and zonal (U) winds at the 850-hPa level, and total precipitation (tp) from the European Centre for Medium-Range Weather Forecasts (ECMWF) fifth generation reanalysis dataset (ERA5) [22] was used for spatial correlation with the $\delta^{18}O$ record and for calculation of anomalies and period differences.

For sea ice data we used monthly sea ice concentration (SIC, 1979–2011) produced using the NASA team algorithm v1.1 (NSIDC-0051) [23], available from the National Snow and Ice Data Center (NSIDC, https://www.nsidc.org, accessed on 2 October 2019).

### 2.4. RACMO Data

We used high-resolution snowmelt and surface mass balance (SMB) data from the Regional Atmospheric Climate Model (RCM) version 2.3p2 (RACMO v. 2.3p2/AP) [24]. The data have a horizontal resolution of approximately 5.5 km × 5.5 km. RACMO v. 2.3 merges the atmospheric model HIRLAM5 with physical package from the ECMWF and is also coupled to a multilayer snow model [25].

### 2.5. Computational Procedure

Before the Pearson's correlation coefficients, anomalies, and climate regime differences are calculated, (1) the seasonal cycle is removed at each grid point by subtracting the corresponding monthly climatology, (2) the data are weighted using the square root of the cosine of latitude to provide equal weighting of equal areas (not performed for the sea ice data), and (3) for the correlations the linear trend is removed.

Anomalies are calculated with the 1979–2011 base period. Differences were calculated by calculating the mean fields of the two periods and subtracting the IPO positive anomaly from the IPO negative (1999–2011 − 1979–1998). The significance of the differences was calculated using two-sample *t*-tests.

Note that we do not investigate Larsen's climate melt dynamics for the more recent periods (2012–present) as it does not overlap with the Palmer $\delta^{18}O$ record.

## 3. Results

### 3.1. Decadal Scale Climate Variability

In this study, we present the $\delta^{18}O$ record from an ice core drilled in Palmer Land, the southern AP. Our results suggest that periods of high ice shelf surface melt are related to the phase of the IPO. Two climate regimes are identified for $\delta^{18}O$ during the satellite era. During the 1979–1998 positive IPO phase and the 1999–2011 IPO negative phase. The different climate regimes associated with these two time periods are presented in Figures 2 and 3. Here we explore the relationship between Palmer $\delta^{18}O$ and climate parameters from the reanalysis data.

*3.2. Ice Shelf Surface Melt Linked to IPO Positive Phase (1979–1998)*

Figures 4a and 5a show the $\delta^{18}$O spatial correlation with Z500 (representing large-scale atmospheric circulation, contours), winds (vectors), and sea ice (shading). The 1979–1998 positive IPO phase is associated with low-pressure systems to the north of the Weddell Sea (Figure 2a,b). This circulation system drives in onshore winds that approach the Larsen shelves from the Weddell Sea area (Figure 2b), resulting in retreat of SIC in the Weddell Sea and an advance in the Bellingshausen Sea/Drake Passage regions (Figure 3a,b). The $\delta^{18}$O correlation plots for the 1979–1998 period are shown in Figure 4. Peaks in $\delta^{18}$O are associated with high-pressure north of the Weddell Sea region, northwesterly winds, SIC reduction in the Bellingshausen Sea/Drake Passage (Figure 4a), and warm air temperatures over the Larsen shelves (Figure 4b).

The climate anomalies (Figures 2 and 3) for this period display opposing signs to the correlation patterns (Figure 4a). For the anomalies (drivers), circulation (Weddell Sea low, Figure 2a,b), wind (high-latitude southeasterly winds approach AP from the Weddell side, Figure 2a,b), and sea ice (increase in Bellingshausen Sea SIC, Figure 3a,b) provide a setting for anomaly negative $\delta^{18}$O values (Figure 4a). The $\delta^{18}$O record that coincides with the satellite era is shown in Figure 4f, the $\delta^{18}$O means of the two IPO intervals are well-separated (two-sample *t*-test, $p = 0.059$).

During this regime when southeasterly winds predominantly approach from the Weddell Sea sector, Palmer $\delta^{18}$O varies in concert with ice shelf melt on the Larsen C. That is the correlation between $\delta^{18}$O, and Larsen C/D surface melt is significant ($r = 0.62$, $p < 0.01$, 1979–1998; Figure 4f). The high-latitude southeasterly winds (Figure 2b) are likely associated with reduced cloud cover conditions that can lead to warming by solar radiation. Our findings are summarized with a schematic (Figure 6a). Figure 6a shows how the circulation and wind anomalies (drivers) and correlation act in opposing directions during IPO positive phase (the wind vectors for the anomalies and correlation point in opposite directions). The southeasterly wind anomalies (offshore) increase the extent of sea ice in the Bellingshausen Sea. A region that coincides with a negative $\delta^{18}$O SIC correlation (Figure 4a). Thus, a Bellingshausen Sea SIC increase is associated with a depletion of the $\delta^{18}$O signal.

The wind belt linked to the (Z500-$\delta^{18}$O correlation) circulation spans a range of latitudes (Figure 4a). Concomitantly to the north of Palmer, the northwesterly winds cross the peninsula and descend over the Larsen ice shelves (Figure 4a). Due to orographic uplifting and rainout, the airflow is dry when it reaches the eastern AP shelves (Figure 4c). In the $\delta^{18}$O correlation patterns, warming on the shelves (Figure 4b) and melt are coincidental with these intense cross-over winds, suggesting that $\delta^{18}$O captures warming induced by dry downwelling winds on the eastern side of the AP, i.e., foehn wind-induced warming and melt during the positive IPO phase (Figure 4b,d).

Note how vast regions of the significant SAT- $\delta^{18}$O correlation coincide with sea ice regions (specifically, significant SIC-$\delta^{18}$O regions). This provides further evidence for a close relationship between $\delta^{18}$O and AP region sea ice variability; as opposed to local Palmer site SAT (traditional interpretation of $\delta^{18}$O) where the $\delta^{18}$O-SAT is not significant (Figure 4b). Porter et al. [15] demonstrated that snow accumulation from the Bruce Plateau ice core is a robust sea ice extent proxy for the Bellingshausen Sea. Similarly, as shown above, the Palmer $\delta^{18}$O displays a consistent negative correlation with Bellingshausen Sea SIC throughout the examined satellite-era period (Figures 4a and 5a). Moreover, sea ice plays an important role in explaining Palmer $\delta^{18}$O variability, more so than local SAT.

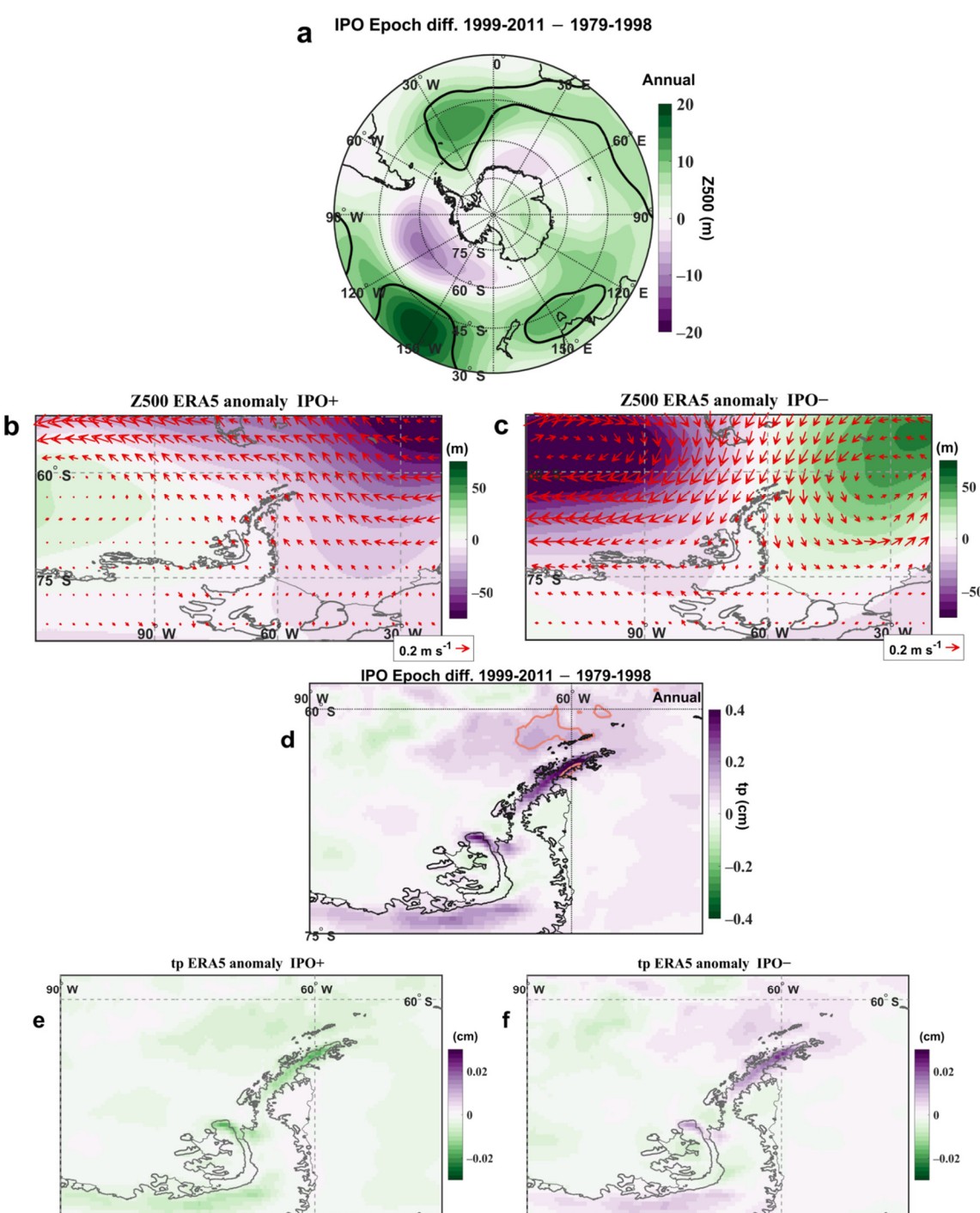

**Figure 2.** Atmospheric circulation, wind, and precipitation anomalies for the 1979–1998 IPO positive and 1999–2011 IPO negative epochs. For annual ERA5, epoch difference (**a**); epoch anomalies (**b**,**c**) geopotential height at the 500-hPa level (Z500, shading) and 850-hPa level winds (vectors of meridional winds (V850) and zonal winds (U850)) and (epoch difference, (**d**); epoch anomalies, (**e**,**f**) for total precipitation (tp). (**b**,**e**) shows the anomalies for the 1979–1998 IPO positive and (**c**,**f**) shows the anomalies for the 1999–2011 IPO negative epoch. Epoch differences (**a**,**d**); 1999–2011 − 1979–1998 provide the *p* < 0.05 significance level contour (two-sample *t*-test) of the difference between the epochs (bold black in a and pink bold contours in (**d**)).

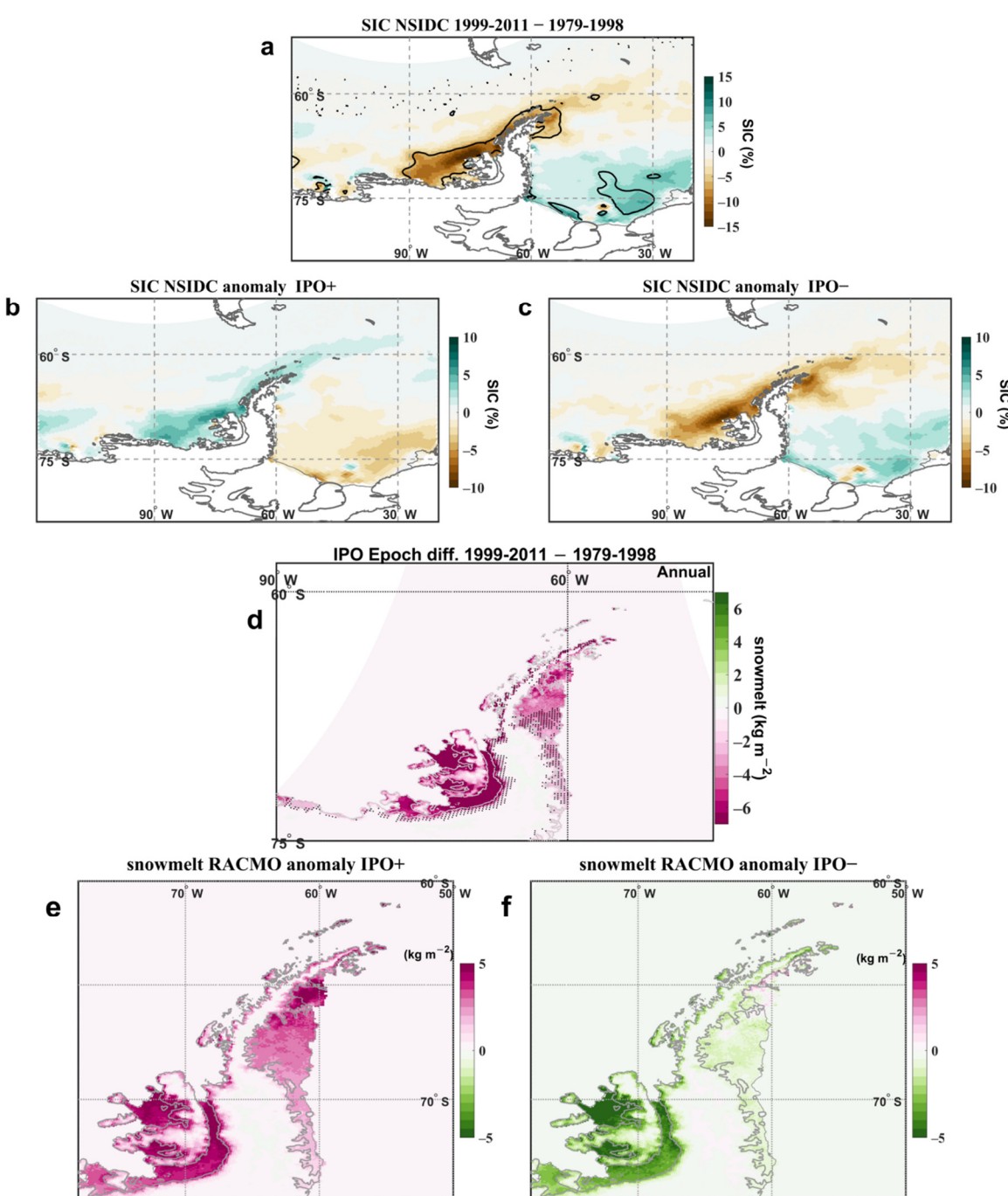

**Figure 3.** Annual NSIDC sea ice concentration, SIC; epoch difference (**a**); epoch anomalies (**b**,**c**) and RACMO snowmelt (epoch difference (**d**); epoch anomalies (**e**,**f**) anomalies. For the IPO positive (**b**,**e**) and negative periods (**c**,**f**). The epoch differences (1999–2011 − 1979–1998) provide the $p < 0.05$ significance level (two-sample $t$-test) of the difference between the epochs (bold black contours in a and black stippling in (**d**)).

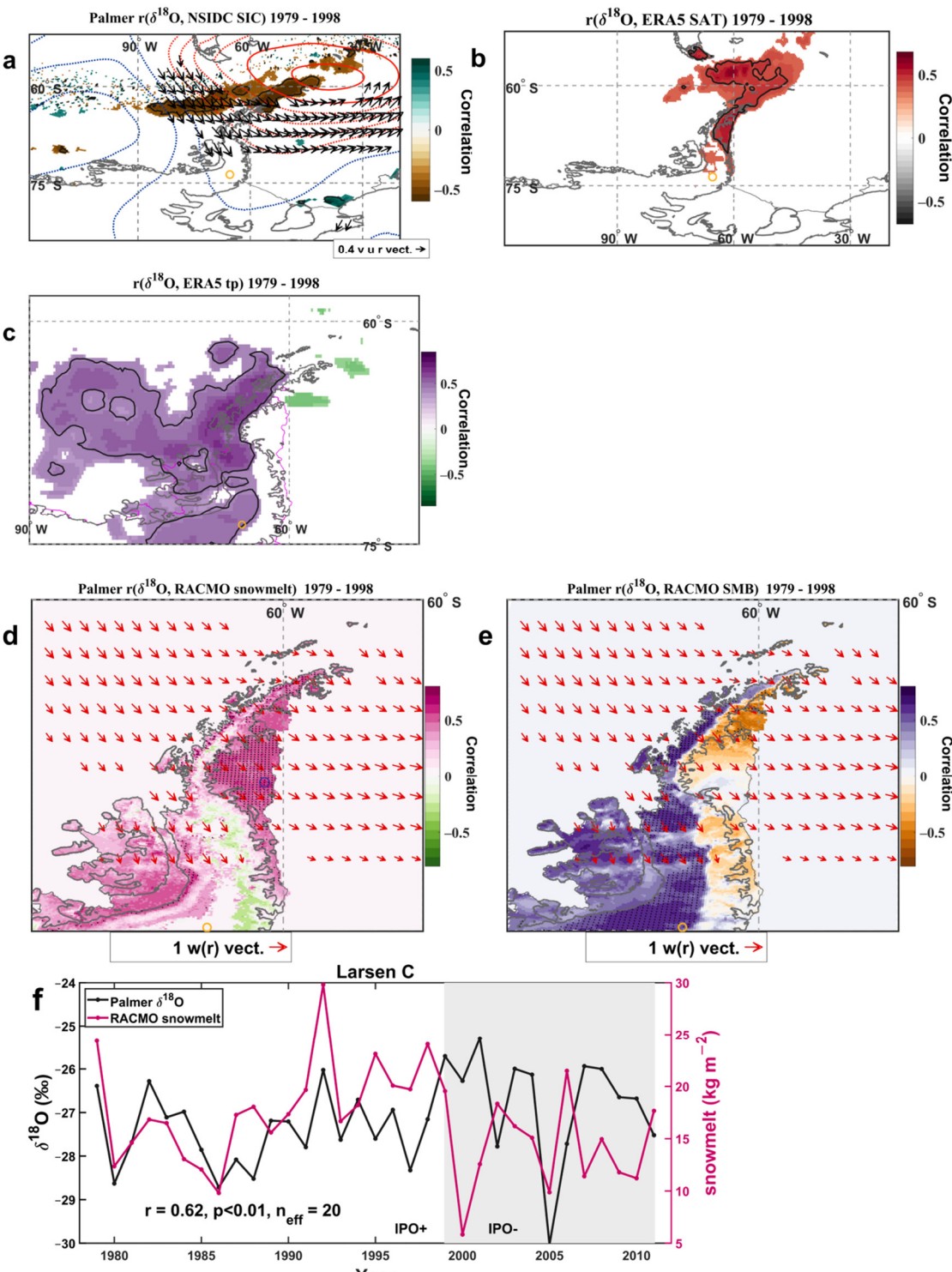

**Figure 4.** δ¹⁸O correlation with reanalysis for the 1979–1998 IPO positive period (**a**) spatial correlation of annual average δ¹⁸O with annual average NSIDC sea ice concentration (SIC, shading ($r$, $p <$ 0.1), black contours ($r$, $p < 0.05$)), with ERA5 geopotential height at the 500-hPa level (Z500, red (positive) and blue (negative) dotted contours shown with $r = 0.1$ increments, starting at $r = 0.1$ and $−0.1$, respectively; bold contours are significant at the $p < 0.05$ confidence level), and with ERA5 winds at the 850-hPa level (V850 and U850, presented as black $r$ vectors of the wind components. At least one of the components is significant at the ($r$, $p < 0.05$) level). (**b**) Spatial correlation of annual average δ¹⁸O with annual average ERA5 surface air temperature (SAT, shading $r$, $p < 0.1$; black contours $r$, $p < 0.05$), and with ERA5 winds at the 850-hPa level (V850 and U850, presented as black $r$ vectors of the wind components. (**c**) Same as b, but for ERA5 total precipitation (tp). (**d**) Spatial correlation of

annual average δ18O with annual average RACMO snowmelt (magenta-green shading, *r*, *p* < 0.1; black stippling *r*, *p* < 0.05; 1979–1998). (**e**) Same as (**d**), but for RACMO surface mass balance (SMB). The Palmer ice core site is indicated by yellow circles in (**a**–**e**). The blue circle in e indicates the location of the most significant correlation with snowmelt. The location from which the snowmelt time series is extracted (shown in (**f**)). (**f**) Time series of annual δ18O (black line) and annual Larsen C snowmelt (magenta line). The effective sample size ($n_{eff}$) is calculated according to [26] (their eq. 31). Autocorrelation is not an issue as $n_{eff}$ is equal to the actual sample size n.

*3.3. Less Melt-Prone Conditions during the 1999–2011 IPO Negative Phase*

The 1999–2011 IPO negative phase is characterized by anomaly patterns of opposing signs compared with the former IPO period, displaying an anti-cyclonic high anomaly over the Southern Ocean, north of the Weddell Sea, with associated meridional winds (from north to south), over the tip and east of the AP (Figure 2c). In contrast to the earlier IPO period, the IPO negative phase climate anomalies display the same signs as their corresponding δ18O correlation patterns. That is, the climate anomalies for this period, circulation (high north of Weddell Sea, Figure 2a,c), winds (northerly, Figure 2a,c), and sea ice (Bellingshausen Sea/Drake Passage SIC decrease, Figure 3a,c), provide a setting for relatively positive δ18O values (Figure 5a,f). The circulation, wind, and sea ice anomalies and δ18O relationship with these parameters for the two IPO phases are summarized in a schematic (Figure 6). Figure 6b shows how the circulation and wind anomalies (drivers) and correlation act in the same direction during this phase (the wind vectors for the anomalies and correlation point in the same direction). The northerly wind anomalies (onshore) reduce the sea ice extent in the Bellingshausen Sea, a region that coincides with a negative δ18O SIC correlation (Figure 5a). Thus, a Bellingshausen Sea SIC reduction is associated with an enriched δ18O signal.

While a positive relationship exists between δ18O and ice shelf melt during the early satellite era, corresponding to IPO positive phase (1979–1998), the link breaks down in subsequent years (Figure 5d). The increase in snowfall at the tip of the AP (Figure 2d–f) is caused by the warmer marine low-latitude origin of the air masses during the IPO negative phase (Figure 2b,c). Moreover, the switch to a moisture-laden air mass regime can also explain the change to a positive correlation between δ18O and snowfall and between δ18O and SMB on the Larsen shelves (Figure 5c,e). There is no significant positive correlation with SAT on the Larsen shelves (Figure 5b), or snowmelt (Figure 5d). The SMB time series, extracted at the correlation maxima on the Larsen D shelf (indicated by the pink circle in Figure 5e), is positively correlated with Palmer δ18O ($r$ = 0.82, $p$ < 0.001). Thus, the increased snowfall at the tip of the AP and the margin between AP and the Larsen shelves during the latter regime (Figure 2d–f) may alleviate the effect of foehn-induced snowmelt.

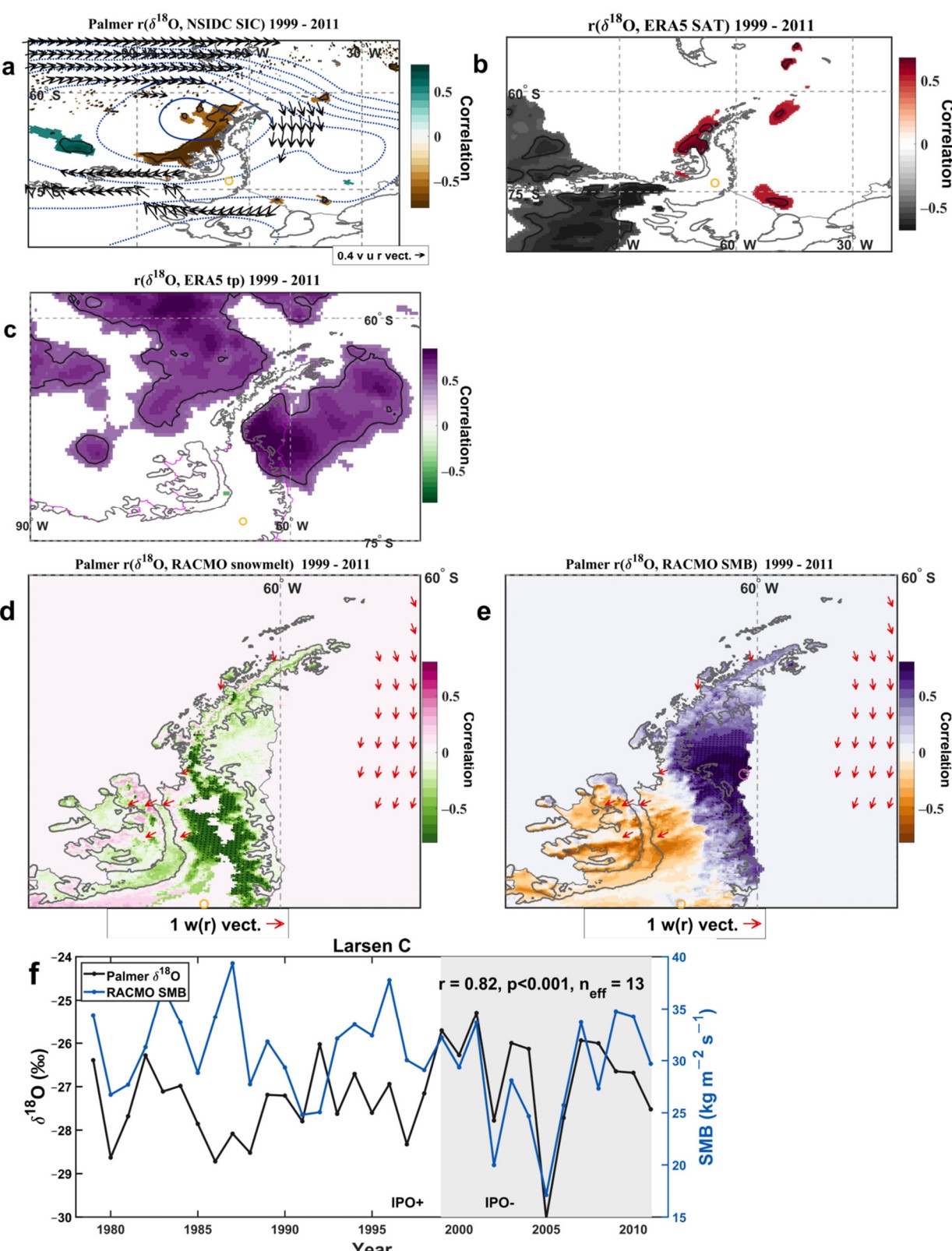

**Figure 5.** δ¹⁸O correlation with reanalysis for the 1999–2011 IPO negative period. (**a**) Spatial correlation of annual average δ¹⁸O with annual average NSIDC sea ice concentration (SIC, shading (*r*, *p* < 0.1), black contours (*r*, *p* < 0.05)), with ERA5 geopotential height at the 500-hPa level (Z500, red (positive) and blue (negative) dotted contours shown with *r* = 0.1 increments, starting at *r* = 0.1 and –0.1, respectively; bold contours are significant at the *p* < 0.05 confidence level), and with ERA5 winds at the 850-hPa level (V850 and U850, presented as black *r* vectors of the wind components. At least one

of the components is significant at the (*r*, *p* < 0.05) level). (**b**) Spatial correlation of annual average δ18O with annual average ERA5 surface air temperature (SAT, shading *r*, *p* < 0.1; black contours *r*, *p* < 0.05), and with ERA5 winds at the 850-hPa level (V850 and U850, presented as black *r* vectors of the wind components). (**c**) Same as b, but for ERA5 total precipitation (tp). (**d**) Spatial correlation of annual average δ18O with annual average RACMO snowmelt (magenta-green shading, *r*, *p* < 0.1; black stippling *r*, *p* < 0.05; 1979–1998). (**e**) Same as d, but for RACMO surface mass balance (SMB). The Palmer ice core site is indicated by yellow circles in (**a–e**). The pink circle in e indicates the location of the most significant correlation with SMB. The location from which the SMB time series is extracted (shown in (**f**)). (**f**) Time series of annual δ18O (black line) and annual Larsen C SMB (blue line). The effective sample size (neff) is calculated according to [26] (their eq. 31). Autocorrelation is not an issue as neff is equal to the actual sample size n.

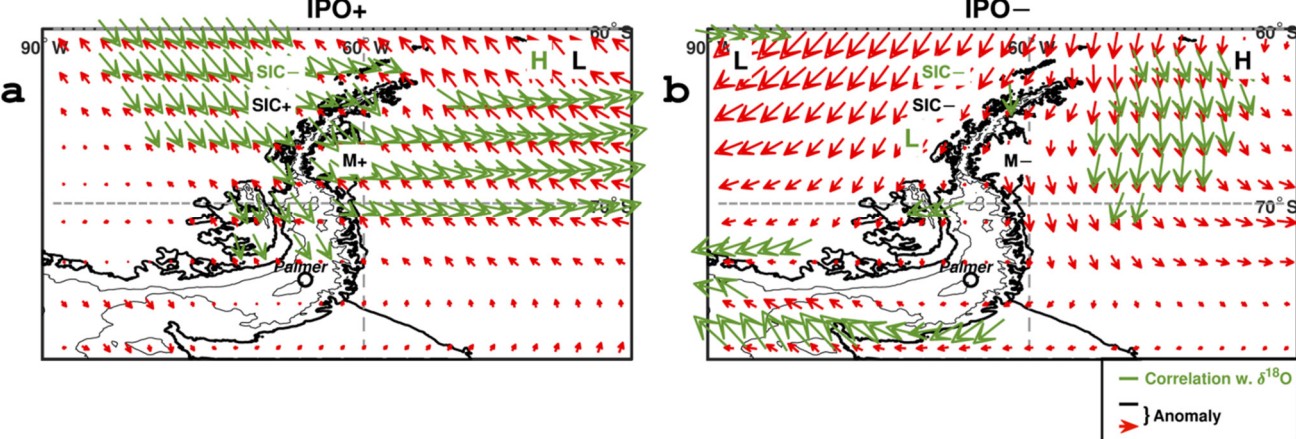

**Figure 6.** Schematics summarizing the results for the IPO (**a**) positive 1979–1998 and (**b**) negative 1999–2011 phase showing the circulation, winds (drivers), and SIC anomalies (black text and red arrows) and correlation (green text and arrows). During the IPO positive phase (**a**) anomalies and correlation oppose. For example, the wind vectors for the anomalies and correlation points in opposite directions. The southeasterly wind anomalies (offshore) expand the sea ice extent in the Bellingshausen Sea. A region that coincides with negative δ18O-SIC correlation. Therefore, a Bellingshausen Sea SIC expansion is associated with a depleted δ18O signal. During the IPO negative phase (**b**) anomalies and correlation align. For example, the wind vectors for the anomalies and correlation point in the same direction. The northerly wind anomalies (onshore) reduce the sea ice extent in the Bellingshausen Sea. A Bellingshausen Sea SIC reduction is associated with an enriched δ18O signal.

## 4. Discussions

Following the collapse of Larsen A and B, Larsen C is at risk of disintegration; especially if we experience a period with a similar climate regime as the 1979–1998 IPO positive phase in the future. The resulting conditions with low-pressure north of the Weddell Sea, and warm air mass instructions may cause accelerated disintegration of the Larsen ice shelves. However, a future positive IPO phase does not necessarily need to correspond with an identical circulation, wind, and melt regime as the one present during the 1979–1998 IPO phase.

Our study shows that the atmospheric circulation during the most recent IPO negative phase is associated with a reduction in the melt on the Larsen C and D ice shelves. Thus, this may have alleviated some of the risk of imminent collapse. Furthermore, the winds have a large meridional component (north-to-south) during the 1999–2011 period, with a band that passes over parts of the AP where the ice shelves have already disintegrated (Figure 2c). However, our research only touches upon atmospheric climate dynamic aspects. For example, winds also interact with ocean currents which in turn can alter ice shelf frontal and basal melt [27]. Furthermore, even though there is a change in snowmelt between the IPO phases, there is no significant change in SMB between the periods (Figure 5f). Thus, any alleviating effect from reduced snowmelt might be canceled by other processes.

Other research has found signs of continual disintegration of the shelves during this period. Including continual surface lowering and thinning of the shelves [28], and surface ponding [7]. Some research indicates that the thinning of the Larsen C is taking place at an accelerated rate during recent decades [29]. However, thinning and surface lowering can result from both firn compaction and basal melt [25], which may not be detected in our purely atmopsheric-focused study.

In climate models, the positive phase of SAM is predicted to continue to intensify, particularly for austral winter [30,31]. In addition to increases in greenhouse gases in summer, the SAM strength is dependent on the stratospheric ozone recovery and the predictions of SAM trends are therefore less certain [31]. A continuation of the positive trend in SAM means that winds become stronger and migrate pole-wards. This can increase the risk of the collapse of ice shelves farther south from foehn wind-induced melt, that is, Larsen C and D.

This research effort demonstrates that one can glean climate dynamics related to ice shelf melting from the $\delta^{18}O$ Palmer record, even though the melt itself is not directly affecting $\delta^{18}O$. Moreover, $\delta^{18}O$ and melt are affected by the same large-scale circulation regimes with associated wind and sea ice changes that affect $\delta^{18}O$ and melt on the Larsen ice shelves. Our research shows that several airmass transport paths can be associated with an enriched $\delta^{18}O$, the different wind correlation vectors for the two IPO periods can be seen in the figures (Figures 4a and 5a).

## 5. Conclusions

Here we present the $\delta^{18}O$ record from the Palmer ice core. The Palmer drill site was located in Palmer Land, right on the divide that separates regions mainly influenced by climate from the Amundsen–Bellingshausen Sea and the Weddell Sea sectors. We demonstrate the importance of IPO on the Palmer $\delta^{18}O$ signal by focusing on two IPO phases during the satellite era.

The 1979–1998 positive IPO phase is associated with anomaly negative $\delta^{18}O$ values. Palmer $\delta^{18}O$ captures the variability of Larsen ice shelf melt during this period. This IPO positive phase was coincidental with the onset of the Larsen ice shelf collapse (Larsen A and Prince Gustav). In contrast, during the IPO negative phase (1999–2011), the climatic setting is less prone to Larsen ice shelf snowmelt; during this regime, the northerly winds passed over the tip and east of the Peninsula and may have acted to preserve the remaining Larsen C ice shelf during the early 21st century.

**Author Contributions:** Conceptualization, B.D.E. and E.R.T.; methodology, B.D.E.; software, B.D.E.; validation, B.D.E. and E.R.T.; formal analysis, B.D.E.; investigation, B.D.E.; resources, E.R.T.; data curation, B.D.E., E.R.T., J.D.H. and D.O.V.; writing—original draft preparation, B.D.E.; writing—review and editing, B.D.E., E.R.T., J.D.H. and D.O.V.; visualization, B.D.E.; supervision, E.R.T.; project administration, E.R.T.; funding acquisition, E.R.T. All authors have read and agreed to the published version of the manuscript.

**Funding:** The ice core drilling and analysis was funded by the British Antarctic Survey, Natural Environment Research Council (NERC, Cambridge, UK), part of UK research and innovation and grant [NE/J020710/1]. Palmer analysis was part funded in collaboration with the Anthropocene Working Group in the assessment of the candidate GSSP-sites. The collaboration was realized in the framework of Haus der Kulturen der Welt's (HKW) long-term initiative Anthropocene Curriculum, an international project for experimental forms of Anthropocene research and education developed by HKW and the Max Planck Institute for the History of Science (MPIWG, Berlin, Germany) since 2013.

**Institutional Review Board Statement:** Not applicable.

**Informed Consent Statement:** Not applicable.

**Data Availability Statement:** The Palmer isotope record (0–133 m below the snow surface, 1621–2011) [32] (https://doi.org/10.5285/cb159b6c-a4b3-4245-80fd-4b4e8122a858, accessed on 1 July 2022) and the Palmer age scale [33] (https://doi.org/10.5285/B3ECA350-79AA-49B2-BD6B-FFEE86AD6559, accessed on 11 February 2022) are available at the UK Polar Data Centre.

**Acknowledgments:** Analysis of the Palmer core was facilitated by the collaborative research project between BAS and the Anthropocene Working Group (AWG) to ratify the stratigraphic Anthropocene. The AWG is coordinating the assessment of candidate GSSP sites in collaboration with the Haus der Kulturen der Welt (HKW, Berlin) in the framework of its long-term project Anthropocene Curriculum. The Anthropocene Curriculum is an international project for experimental forms of Anthropocene research and education developed by HKW and the Max Planck Institute for the History of Science (MPIWG, Berlin) since 2013. We acknowledge Julius Rix and Catrin Thomas for drilling and field support and Shaun Miller and Emily Ludlow for laboratory support.

**Conflicts of Interest:** The authors have no conflict of interest to declare.

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
