# Peer review of "Decadal Scale Variability of Larsen Ice Shelf Melt Captured by Antarctic Peninsula Ice Core"

_geosciences, doi:10.3390/geosciences12090344_

Round 1
Reviewer 1 Report
Review of “Decadal scale variability of Larsen ice shelf melt captured by Antarctic Peninsula ice core.” by Emanuelsson et al.
Emanuelsson et al. provide a succinct overview of the influence of the Interdecadal Pacific Oscillation on the climate of the Antarctic Peninsula over the post-1979 satellite era. The authors compare the stable water isotopes from the Palmer core to various, regional atmospheric and cryospheric parameters which highlights non-linearities in preserved ice core signals due in part to the decadal-scale variability and large-scale climate oscillations or patterns. The authors make reasonable conclusions, and the results appear valid. Although I do believe some of the results could be further clarified, I recommend that the paper is accepted with minor revisions.
Major Revisions
In reading this paper, I struggled with comprehending all of the moving parts, so to speak. I would recommend adding a simplified schematic to sum up the various dynamics during both a positive IPO and negative IPO. This could be a separate figure 6, or they could be panels in Figures 4 and 5, respectively, if space is an issue.
I suggest a schematic because I think I found a disparity, but I could also be completely misinterpreting the relationships. In the positive IPO scenario, higher snowmelt is observed across much of the peninsula and ice shelves (Fig. 3e). The relationship between snowmelt over the Larsen shelf and Palmer oxygen isotopes is positive (Fig. 4d). However, this is a setting for lower isotope anomalies which would suggest lower snowmelt, correct?
Also, how do your results compare with those from the Bruce Plateau core [Goodwin et al. 2016; Porter et al., 2016]? They also noted non-linearities in response to Pacific decadal variability and the relationship to regional sea ice.
Minor Revisions
Figures 2 & 3: I would mark where the Palmer ice core site is.
Figures 4 & 5: I would note that the circles indicate the Palmer site and the snowmelt/SMB from Larsen in the caption.
Line 64: I would move this sentence to line 79.
Line 99: I agree that linear trends are best removed in this type of analysis. However, I am curious how strong the trends were given your discussion that the negative IPO may have mitigated melt in the region.
Line 106: It feels a tad odd to discuss results before showing them. I wonder if it might be more organized to perhaps discuss the previous literature on the IPO influence to justify why you are splitting the time series rather than your own results justifying your methodology.
Lines 121-124: I struggled here to determine your meaning. There is a typo or two as well, but I think you may want to clarify when you discuss opposing signs. This sounds a bit as if there is a contradiction to your results rather than just noting the relationship with lower isotopes. Does that make sense? Please let me know if that’s unclear or if I have completely misinterpreted this section.
Line 123: Are the averages in the stable isotopes significantly different between the two periods?
Line 127: Am I able to see the high-latitude winds in a figure or am I meant to just glean this from basic climatology? If the former, I would add a figure reference. If the latter, I would add a literature reference.
Lines 203: It seems your results are supporting that the sea ice around the peninsula is dynamically driven by the winds and less so by temperature. Am I correct there? Does the previous literature suggest this as well? I am genuinely ignorant here and just curious. The next paragraph seems to indicate that there is a temperature element as well.
Lines 219-225: How can the snowmelt or surface mass balance be reconstructed with all of these non-linearities? Your results certainly show that assuming the same relationship back in time can be problematic. Could you use one of the many IPO reconstructions to inform the relationship you are modeling? Not for this manuscript, but perhaps in the future. Of course, there are many dissimilarities in those IPO reconstructions. Gosh, what an interesting puzzle though!
References
Goodwin, B. P., E. Mosley-Thompson, A. B. Wilson, S. E. Porter, and M. Roxana Sierra-Hernández. 2016. Accumulation variability in the Antarctic Peninsula: The role of large-scale atmospheric oscillations and their interactions. J. Clim., 29(4), 2579-2596, doi:10.1175/JCLI-D-15-0354.1
Porter, S. E., C. L. Parkinson, and E. Mosley-Thompson. 2016. Bellingshausen Sea ice extent recorded in an Antarctic Peninsula ice core. J. Geophys. Res. Atmos., 121, 13,886–13, 900, doi:10.1002/2016JD025626
Reviewer 2 Report
This article describes a study of how the variability of the stable water isotope record from an Antarctic Peninsula ice core (Palmer) that overlaps with the instrumental record can be used to derive information about the surface melt on the Larsen Ice Shelves C and D are affected by the same large scale climate regimes and show similar patterns of decadal scale variability, showing that the IPO positive phase between 1979-1988 corresponds to conditions more favourable to surface melt and negative anomaly isotope values. Conversely, the IPO negative phase of 1999-2011 show conditions less favourable to surface melt and positive del18O anomaly values. This is an interesting study that can provide a larger scale climate context to the collapse of the Larsen A and B ice shelves and likelihood of collapse of other vulnerable ice shelves on the Antarctic Peninsula.
The paper is generally well written with good figures and appropriate referencing. However, I found the figures quite difficult to follow, being quite detailed, but far from the main text where only very brief explanations are found. Eg, Figs 4a and 5a are very detailed, and would warrant much more explanatory text.
There also appears to be some layout issues making it confusing which captions belong to which figures. This could be improved by splitting the figures up so that there are less subparts, keeping their captions closer. In some cases the captions are simply too brief to be able to follow.
The computational procedure section needs much more detail. For example, is it the Pearson correlation which is used, and how was n_eff calculated, and why (to account for autocorrelation?)
I also would have appreciated some explanation about why the monthly climatology was removed when the annual menas of the reanalysis data were used. Presumably the ice core record values are also annual means, but this is also not clear and how the "year" is defined, eg Jan-Dec?
Where IPO epoch differences are calculated (Fig 2a,d), please provide details on how this calculation actually done.
The main criticism I have for this article though is that while there are interesting relationships that can be found between the ice core data and the climate reanalysis and model data, it is not clear what the ice core data itself adds to the story. Presumably the next step would be to do a reconstruction of some of these climate parameters, or the IPO index itself to examine how likely melt was in past IPO positive phases, putting the more recent epochs into a longer context?
The discussion section does start to outline the limitations of the relationships found to explain or predict possible ice shelf collapse, but could be expanded upon.
Specific comments:
line 105: "Here we present the del18O record..." - the record itself is not really presented anywhere in the paper by itself with any explanation of its properties or features. The time series is only presented where it is correlated with RACMO data.
line 113-114: Not a complete sentence
line 118-120: Also not a complete sentence
line 123: Is there something missing here, or is the full stop misplace?
line 190: "a-e figure details are the same as Fig. 4" That may be so, but Fig 4 is a long way away and also complicated, please add the relevant information in this caption.
line 252-254: please provide DOIs for the datasets.
